# Real-Time PCR (qtPCR) to Discover the Fate of Plant Growth-Promoting Rhizobacteria (PGPR) in Agricultural Soils

**DOI:** 10.3390/microorganisms12051002

**Published:** 2024-05-16

**Authors:** Ilenia Iosa, Caterina Agrimonti, Nelson Marmiroli

**Affiliations:** Department of Chemistry, Life Sciences and Environmental Sustainability, University of Parma, A-43124 Parma, Italy; ilenia.iosa@unipr.it (I.I.); nelson.marmiroli@unipr.it (N.M.)

**Keywords:** microbial consortia (MC), plant growth-promoting rhizobacteria (PGPR), real-time PCR (qtPCR), traceability, biostimulants, sustainable agriculture

## Abstract

To optimize the application of plant growth-promoting rhizobacteria (PGPR) in field trials, tracking methods are needed to assess their shelf life and to determine the elements affecting their effectiveness and their interactions with plants and native soil microbiota. This work developed a real-time PCR (qtPCR) method which traces and quantifies bacteria when added as microbial consortia, including five PGPR species: *Burkholderia ambifaria*, *Bacillus amyloliquefaciens*, *Azotobacter chroococcum*, *Pseudomonas fluorescens*, and *Rahnella aquatilis.* Through a literature search and in silico sequence analyses, a set of primer pairs which selectively tag three bacterial species (*B. ambifaria*, *B. amyloliquefaciens* and *R. aquatilis*) was retrieved. The primers were used to trace these microbial species in a field trial in which the consortium was tested as a biostimulant on two wheat varieties, in combination with biochar and the mycorrhizal fungus *Rhizophagus intraradices*. The qtPCR assay demonstrated that the targeted bacteria had colonized and grown into the soil, reaching a maximum of growth between 15 and 20 days after inoculum. The results also showed biochar had a positive effect on PGPR growth. In conclusion, qtPCR was once more an effective method to trace the fate of supplied bacterial species in the consortium when used as a cargo system for their delivery.

## 1. Introduction

In recent decades, to fulfill the need for enhanced sustainability in agriculture, the search for biotechnological products as green alternatives to chemical fertilizers, pesticides, and herbicides has intensified. The utilization of new-generation plant biostimulants can reduce dependency on synthetic agrochemicals, thus contributing to mitigating the negative environmental and human health-related impacts [1,2,3]. Microbial-based biostimulants have received increasing attention for their plant growth promotion, for biocontrol activities [2,4], and for preserving soil health and biodiversity [3]. Microbial biostimulants are usually based on plant growth-promoting microorganisms (PGPMs), namely, bacteria, fungi, and microalgae [5]. In particular, the use of the subgroup plant growth-promoting rhizobacteria (PGPR) is becoming frequent in agricultural applications [6]. PGPR colonize the rhizosphere of the plants while directly or indirectly improving the uptake of nutrients, plant resistance to both abiotic/biotic stresses, soil fertility, crop yield, and the overall quality of the products [7].

Despite their great potential, the applicability of PGPR in the field remains a challenge because of the variability and complexity of the environmental conditions, which make a robust and reproducible evaluation of the effects on the crops difficult [8]. Many factors can, positively or negatively, affect the persistence and the efficacy of PGPR after their inoculation into the soil [4]. As reviewed by Rilling et al. [9], more relevant factors include the biology and chemistry of the soil, plant species and phenological stages, as well as the quantity and composition of the root exudates. PGPR’s establishment in the soil may also be altered by competition with the indigenous microbiota, especially if the inoculum’s concentrations are not sufficient to allow efficient colonization [2]. Natural microorganisms’ dispersion, attributable to rain, dust, or movement by invertebrates, must be taken into account [10]. Considering all these variables, ideally, the efficacy of a given microbial biostimulant should be tested over different climatic and geographical conditions and for soils with different physical–chemical properties to assess its degree of adaptability to environmental changes [11]. In field conditions, it becomes fundamental to understand their shelf-life and their interactions with the host plant and with the soil microbiota in order to bridge the gap between laboratory results and field performance [4,12].

Since biotic and abiotic factors can affect microbial persistence in the soil or decrease their functional efficiency, the association of PGPR with different functions in microbial consortia (MC) is more effective than the use of single species. Also, synergism among microorganisms may reinforce their action by sharing nutrients, removing inhibitory products, stimulating beneficial physiologic traits, and replacing those strains which are unable to establish in the soil. Moreover, a high genetic diversity within MC improves their resilience and their capacity to counteract the adverse environmental conditions [13]. But microorganisms’ species are not always compatible. Therefore, the development of MC requires the search for microorganisms with complementary functions, but limited competition.

After literature screening and microbial compatibility tests, Tabacchioni et al. [14] developed three MC, designed as MC_A, MC_B, and MC_C, including bacterial species with functions such as nitrogen fixation, phosphorus solubilization (*Azotobacter chroococcum/vinelandii*, *Paraburkholderia tropica*), amylolytic activity (*Bacillus amyloliquefaciens*), production of auxin or auxin-related compounds (*Bacillus licheniformis*), plant growth-promoting effects (*Burkholderia ambifaria*, *Komatagaella pastoris*, *Pseudomonas granadensis*, *Trichoderma harzianum*, *Rahnella aquatilis*), and biocontrol of pathogens (*Bacillus* spp., *Pseudomonas fluorescens*) [15,16]. The performance of PGPR strains, taken as singularities or as consortia, are improved by combining them with mycorrhizal fungi, such as the arbuscular fungus *Rhizophagus intraradices,* to protect plants from excess of salinity, lack of water, and presence of phytotoxic elements, as well as to increase plant nutrient uptake [17,18]. But also, some amendments can exert a positive effect on soil and plants: For example, biochar, obtained by the pyrolysis or pyrogasification of biomasses, stimulates soil microbial activity, improves nutrient cycling, and reduces nutrient leaching [19,20].

Beneficial effects were also found after pot experiments that evidenced an increase in the shoot and root biomass of wheat and maize plants and in the biodiversity of the soil’s indigenous rhizo-microbiota [21]. Furthermore, treatment with these MC upregulated genes involved in amino acid metabolism, glycolysis/gluconeogenesis, secondary metabolite biogenesis, and lipid metabolism in wheat and maize. In maize, genes involved in photosynthesis and starch metabolism were also upregulated [21]. The effects were particularly pronounced when employing MC_C in combination with biochar and *Rhizophagus intraradices.*

To understand the interactions of PGPR with plants and soil, reliable methods of detection and quantification of microbial cell/DNA is necessary. Quantitative real-time PCR (qtPCR) is fast, does not require bacterial cultivation, and may reach a limit of detection around 10^2^ CFU/g of soil [9]; therefore, it has been used in many applications, for example, food microbiology [22,23]. Moreover, the innovation of chemicals of qtPCR-like plasmonic nanoparticles [24] and/or the development of low-cost, hands-on qtPCR [25] demonstrate that this technology is constantly evolving and easily adaptable, even outside the laboratories. Therefore, the integration of qtPCR with other methods, like reporter gene analysis, immune-associated techniques, and next-generation sequencing (NGS), can provide a picture of the behavior of PGPR in the soil.

This work reports on the development of a qtPCR protocol based on the use of the intercalating dye SYBR Green [26] to tag microorganisms of the MC_C consortium, as mentioned above, in soil samples. The samples were collected from an experimental field in which the pot experiments conditions mentioned in ref. [21] were tested in a real agronomic and environmental context. The work was conducted through the following steps: (i) identification of DNA sequences specific to the strains of the consortium; (ii) screening of primer pairs to produce unique amplicons from each single bacterial DNA of MC_C; and (iii) testing the primer on soil samples sown with two varieties of wheat, Bramante and Svevo, collected at different times after the bacterial inoculation.

## 2. Materials and Methods

### 2.1. Microbial Strains, Microbial Consortia, and Biochar Production

The consortium MC_C, made out of five different bacterial strains that show growth-promoting activities, as indicated in Table 1, was provided by CCS Aosta S.r.l (Aosta, Italy) in a lyophilized formulation. In the consortium, each of the bacterial strains was stabilized with sterile micronized zeolite [16].

The five tester microbial strains were provided in stabbed form by the National Agency for New Technologies, Energy and Sustainable Economic Development (ENEA, Rome, Italy). The tester bacterial strains were grown overnight in 2 mL of LB (Luria–Bertani) liquid medium (Tryptone 10 g L^−1^; yeast extract 5 g L^−1^; NaCl 10 g·L^−1^) at 28 °C. Liquid suspensions were centrifugated at 3000× *g* for 5 min, and the pellets were collected and stored at −20 °C for DNA extraction for primer testing and standard curves.

The arbuscular mycorrhiza fungus (AMF) was purchased from the MycAgro lab (Bretenière, France). The granular inoculum consisted of *R*. *intraradices* propagules (spores, hyphae pieces) and mycorrhizal root pieces (10 propagules g^−1^ containing a mix of spores, mycelium, and mycorrhizal root pieces) homogenized with solid particles of clay and zeolite mineral [21]. Biochar was produced by Iridenergy (Parma, Italy) from wood pellets; its properties have been described by Marmiroli et al. [19].

### 2.2. Experimental Set up of the Wheat Field Trial

The wheat field experiment started in November 2022 at the agricultural experimental farmPodere Stuard (Parma, Italy); https://www.stuard.it/ (URL accessed on 14 May 2024). Seeds of wheat cultivars *Triticum durum* (cv. Svevo) and *Triticum aestivum* (cv. Bramante), released by Produttori Sementi Bologna PSB s.p.a. (Bologna, Italy), were sown at a density of 400 seeds per square meter (m^−2^). In this field trial, MC_C was added to the soil in combination with biochar and/or AMF. MC_C was added to the soil at a ratio of 2:1 *W*/*W* (powder/seeds), while AMF was applied at 1:1 *W*/*W* (granules/seeds). The experiment was set up in a split plot design, with plots of 3 m^2^ per treatment. The biochar was mixed with the soil before sowing at 200 g m^−2^.

After 119 days from sowing, a second addition of MC_C was spread near the roots of the plants in the same amounts as previously. Meteorological parameters were collected daily by an automatic computerized weather station installed close to the experimental field (San Pancrazio, Parma). During the sampling period (22 March 2024–5 May 2023), the total rainfall was 72.9 mm, and the mean air temperature registered at 12.7 °C (min 6.1 °C–max 16.6 °C) (Figure 1).

### 2.3. Soil Sample Collection

Samples were collected in the following conditions by taking about 10 g of soil at a depth of about 3 cm as close as possible to the roots: (i) control, without any treatments; (ii) AMF + MC_C; and (iii) AMF + biochar + MC_C. Rhizospheric soil samples were collected in November 2022, six days after wheat sowing, upon sprout emergence, and in the spring of 2023 following the addition of MC_C to the seedlings and plants, on the dates 30 March 30 (T1), 6 April (T2), 17 April (T3), and 5 May (T4). For each condition and time, two separate pools of three plants were collected.

The pH of all the soils was in the range of 6.6–7.5, and the soil electrical conductivity (EC) was in the range of 110.2–450.4 μS·cm^−1^. The samples were stored at −20 °C until DNA extraction.

### 2.4. DNA Extraction and Quantification

DNA was extracted from (i) pellets of tester strains obtained as described in 2.1 and (ii) 1 g of rhizospheric soil samples using the NucleoSpin Soil kit (Macherey-Nagel, Dueren, Germany), following the manufacturer’s instructions.

DNA concentration was determined spectrophotometrically with a Varian Cary^®^ 50 Bio UV-Visible Spectrophotometer (Agilent Technologies, Santa Clara, CA, USA) by measuring at the two wavelengths of 260 nm and 280 nm. The ratio at 260 nm/280 nm was used to estimate the purity of the extracted DNA.

### 2.5. Searching Primers Specific for Microbial Species of MC_C

The search for primer pairs search followed two criteria: (i) the scientific literature, to find primers already available for the targeted bacterial species; (ii) in silico analyses, to compare the entire genomes of the five microbial species of MC_C (BLAST algorithm available at https://blast.ncbi.nlm.nih.gov/Blast.cgi; (URL accessed on 14 May 2024) or specific genes’ sequences (MegaX software vs 11) [27] available at https://www.megasoftware.net (URL accessed on 14 May 2024). Primers were designed using the Primer3 Plus web tool (vs 3.3.0), available at https://www.primer3plus.com/index.html (URL accessed on 14 May 2024). The Raq4 primer pair was designed based on the genomic sequence of *Rahnella aquatilis* HX2, coding for the F0F1 ATP synthase subunit beta (CP003403.1:4937125-4938507).

To estimate the copy number of each target sequence, primer sequences were aligned against those of each bacterial genome using the BLAST algorithm. Since the pairing sites were unique in at least ten strains of each species, we concluded that one copy of the amplicon corresponded to a single genome equivalent (GE). The GEs were calculated based on the length of the reference genome of each bacterial species using the following equation [28]:$$No.of\;GE\;copies=\frac{\left(6.02\times 10^{23}\;copies\cdot {mol}^{-1}\right)\times DNA\;amount\;(g)}{Genome\;lenght\left(bp\right)\times \left(660g\cdot {mol}^{-1}\cdot {bp}^{-1}\right)}$$

### 2.6. Real-Time PCR

Real-time PCR was conducted using the CFX96 apparatus (Biorad, Hercules, CA, USA). An aliquot of 1 ng of microbial DNA, corresponding approximately to 10^5^ GEs, 5 ng of MC_C DNA, or 40 ng of DNA extracted from rhizospheric soil, was amplified in 10 µL of mix containing 1× POWRUP SybrGreen Master Mix (Thermofisher, Waltham, MA, USA) and 250 nmol L^−1^ primers forward (FW) and reverse (RV). The cycling program included a 3 min preincubation at 95 °C, followed by 40 cycles at 95 °C for 10 s and at 62 °C for 30 s. The amplicon dissociation was conducted in the temperature range of 65–95 °C, with an increment of 0.1 °C·s^−1^. Data were analyzed with CFX Maestro^TM^ Software vs 2.2 (Biorad).

The inhibitory effects of soil DNA were assessed by comparing the reaction efficiency (E) and the coefficient of determination (R^2^) of standard curves in the presence or absence of 15–40 ng of DNA extracted from the AMF + biochar sample. Because this DNA did not significantly affect the E or R^2^ of the standard curves, the conclusion was that there were no inhibitory effects, even with 40 ng of DNA. Therefore, this amount was used for all reactions.

The standard curves constructed with the five serial dilutions of bacterial genomic DNA, ranging from 1 to 10^−4^ ng and corresponding to 10^5^–10^1^ GEs, were included in each reaction made with three replicates for dilution. The results were used to interpolate the quantity of the target DNA in the sample.

### 2.7. Data Analysis

ANOVA comparisons were aided by Daniel’s XL Toolbox for Excel, version 7.3.2, by Daniel Kraus, Würzburg, Germany, available at https://www.xltoolbox.net (URL accessed on 14 May 2024); box plots were designed with R software, version 4.3.1.

## 3. Results

### 3.1. Screening of Primers for Specific Amplification of Strains 

A total of 90 primer pairs, either taken from the literature or designed in our laboratory, were tested using qtPCR conducted on DNA extracted from liquid cultures of each of the bacterial strains of MC_C. As a result, 63.3% of these primers did not give any amplification. Other primers, about 30%, amplified DNA from more than one bacterial species; therefore, they were not specific. Finally, three primer pairs specific for *B. ambifaria* (Bamb1196, Bamb3350, Bamb4475), one for *B. amyloliquefaciens* (rpsj), and one for *R. aquatilis* (Raq4) were found (Table 2).

### 3.2. Qualitative Test on DNA Extracted from Soil

Preliminary tests were conducted on soil samples mixed with MC_C powder at the percentages of 10%, 5%, and 2.5% (*W*/*W*) to assess the efficacy of primers in detecting target bacteria. Possible inhibitory effects of soil compounds and humic or fulvic acids released during the DNA extraction, as well as the presence of similar soil microorganisms that could interfere with amplification, were considered. The primers Raq4, rpsj, Bamb1196, Bamb4475, and Bamb3350 easily detected their target microorganisms, with a threshold cycle (CT) ≤ 24. The primer pair Bamb3350 was chosen to tag *B. ambifaria* in the further experiments, because the CT values were lower as compared to those obtained with the other two pairs.

The primers were then tested on DNA extracted from soil collected near the seedlings of Svevo cultivar, supplemented with AMF + MC_C or AMF + biochar + MC_C taken six days after sowing. Figure 2 shows how all primer pairs detected their target microorganisms in the samples supplemented with MC_C, while no amplification was found in non-supplemented controls, except for the rpsj primer pair, but only after a considerable number of amplification cycles.

### 3.3. Quantification of DNA from Microbial Species in Soil Samples

qtPCR conducted on rhizospheric samples collected in the spring of 2023, at different times after the supplement of MC_C, evidenced a general trend of growth for the bacteria *B. ambifaria*, *B. amyloliquefaciens,* and *R. aquatilis* (Figure 3*)*.

A small amplification was observed in the control samples, but much lower than in the supplemented samples. This was certainly due to the presence of autochthonous bacteria similar to those targeted. However, the difference between the control and supplemented samples was significant, as shown in Table 3, which displays all statistical comparisons among the samples.

Even though the targeted bacterial species showed a similar trend of growth after the inoculum, differences among them were observed. For the cv. Bramante, in the presence of AMF + MC_C and AMF + biochar + MC_C, *B. ambifaria* grew steadily for up to 26 days after MC_C supplementation, declining sharply after 44 days (T4). For the supplement of AMF + MC_C, a maximum average value of GEs corresponding to 7.32 × 10^5^ was reached at T2, 15 days after addition. Instead, for the AMF + biochar + MC_C supplement, the maximum average value of GEs corresponding to 1.90 × 10^6^ was observed at T2, 15 days after MC_C addition. In the interval T2–T3, a higher number of GEs was observed in the presence of AMF + biochar + MC_C (T2 = 1.90 × 10^6^; T3 = 1.64 × 10^6^) with respect to those reported for AMF + MC_C (T2 = 6.36 × 10^5^; T3 = 7.32 × 10^5^), although this was not significant (*p* = 0.66514). A similar trend was observed for the cv. Svevo, with an exceptional decrease in GEs at T2 that can be considered accidental. In the Svevo cultivar, the combination of AMF + biochar + MC_C significantly increased the growth of *B. ambifaria* with respect to AMF + MC_C (*p* = 0.00730).

*B. amyloliquefaciens,* in the presence of cv. Bramante and supplemented with AMF+biochar, showed a growing trend similar to *B. ambifaria*, with a maximum average value of GEs at T2 and T3 of 3.63 × 10^5^ and 3.09 × 10^5^, respectively. These GE values were lower as compared to those observed for the bacterium *B*. *ambifaria* at the same sampling times (T2 = 1.90 × 10^6^; T3 = 1.64 × 10^6^). A comparison of the entire set of data did not show significant differences between AMF + MC_C and AMF + biochar + MC_C (*p* = 0.05269). The differences became significant when comparing the GEs recorded at T2, T3, and T4 (*p* = 0.00502). For the cv. Svevo, growth was rather limited, with a maximum average value of GEs of 3.63 × 10^5^ obtained for the AMF + biochar + MC_C condition at T2. However, the GEs recorded for AMF + MC_C and AMF + biochar + MC_C were significantly higher than those recorded for the control (*p* = 0.00170 and 2.7 × 10^5^, respectively). In the same cultivar, a significant difference between GEs detected in the presence of AMF + MC_C and AMF + biochar + MC_C was observed, either comparing all sampling times or T2, T3, and T4 only (*p* = 0.00239 and 0.00553, respectively).

When supplemented to cv. Bramante, *R. aquatilis* reached a maximum of growth at T3 in the presence of AMF + MC_C, with an average value of GEs of 1.72 × 10^4^, even though the growth was limited, but the GEs were significantly higher as compared to the control *(p* = 0.04419). In this case, the differences between the conditions AMF + MC_C and AMF + biochar + MC_C were not significant (*p* = 0.62860) when comparing T2, T3, and T4 only (*p* = 0.77247). In the cv. Svevo, GEs in the presence of AMF + biochar + MC_C were significantly higher than in the presence of AMF + MC_C when comparing all times and T2, T3, and T4 (*p* = 0.00247 and 0.00282, respectively). Since the GE number in the Svevo sample collected at time T2 was more than 100-fold higher than in the other samples, it was considered an outlier and was not included in subsequent analyses. The differences, in this case, were not significant (*p* = 0.08138), but GEs in the presence of AMF + biochar + MC_C were slightly higher than in the presence of AMF + MC_C, as revealed by the average and median ratio between these two conditions (1.09839 and 1.13917, respectively) (Table 4). This trend was shown for almost all conditions, confirming the general positive effect of biochar on MC_C bacterial growth (Table 4).

## 4. Discussion

The goal of this study was to design a qtPCR protocol for the species-specific quantification of the PGPR within the consortium MC_C, as previously described [14]. Among the techniques currently available for bacterial tracking in environmental samples, qtPCR combines high environmental safety, reproducibility, sensitivity, and specificity, allowing for the rapid evaluation of large-scale field experiments, differently from traditional culturing methods [9,31]. Identification and quantification based on these methods are limited by the fact that not all microorganisms are cultivable. Moreover, culturing methods are laborious and time-consuming, and their application can be complicated by the presence of native microorganisms which are morphologically similar to the targeted bacteria in soils [32]. The qtPCR system overcomes these limitations because it detects only specific DNA sequences [33]. Other methods based on sequence annotation are available, but they are more costly and not truly quantitative [34]. For these reasons, qtPCR was successfully used to detect and enumerate bacterial populations of single or multiple coinoculated PGPR strains in plant tissues and in rhizospheric soil [31,32,35,36,37,38]. However, to develop a reliable and feasible detection method, a deep understanding of the genome sequence of the target microorganism is needed [39]. Herein, a search for species-specific primers is reported, which resulted to be a difficult task due to the lack of sequencing data for the strains used in the field trials. After the screening of one hundred primer pairs, three resulted to be suitable to trace the PGPR bacteria *R. aquatilis*, *B. ambifaria,* and *B. amyloliquefaciens.* Some studies have reported strict relationships between the genera *Azotobacter* and *Pseudomonas,* and, effectively, the majority of the primers tested amplified the DNAs of both *P. fluorescens* and *A. chroococcum* [40,41]. Moreover, some strains of *P. fluorescens* are subjected to extensive recombination through mobile elements, and this may increase the variability within the species [42]. For this purpose, the sequencing of the specific strains used in this study will be of great help to complete the panel of MC_C, bypassing strain differences.

The species-specific primers designed/found were used to monitor the abundance and shelf lives of the strains of MC_C added to the wheat cultivars Bramante and Svevo in field experiments.

It is recognized that sampling of rhizospheric soil represents the most critical step in the entire tracking process to ensure the quality and robustness of the data [43]. Uneven distribution of the inoculated microorganisms in the soil, occurrence of bacteria’s dispersion phenomena, and the impact of climatic factors can affect replicates’ reproducibility, as shown by the wide distribution of data for some samples and by the presence of some outliers. Moreover, the presence of native microorganisms closely related to the bacterial target can make the detection difficult, leading to cross-amplification [31]. Despite the difficulties, this work has demonstrated that three out of the five bacterial species included in MC_C are capable of colonizing and growing into soils sowed with two wheat cultivars for up to a month, even in rather unfavorable climatic conditions. A general trend of the growth of the three microbial species was found: GE values increased for 2–3 weeks after the inoculum and then dropped sharply. As indicated by meteorological data, the consortium MC_C was added to the soil at the end of winter, when temperatures were around 10 °C, which was not optimal for these mesophilic bacteria. During the period between T2 and T3 sampling, the temperatures dropped further to 5–6 °C in conjunction with low rainfall and a relative drought condition. However, the climatic conditions do not seem to have had a particular effect on the microbial growth, considering that GEs dropped out between T3 and T4 despite the increase in temperature and rainfall in this period.

Differences in the dynamics of growth among the three species should be related to the genetic and physiological characteristics which influenced their colonization and persistence in the soil [33]. *Burkholderia* species are characterized by a large phenotypic plasticity and by a capacity to endure environmental stressors such as harsh temperature variations [44]. Although their optimal growth temperature is 37 °C, some strains of the *Burkholderia epacian* group, which includes *B. ambifaria*, can grow in extreme environments, like distilled water at 10 °C [45]. This evidence could explain why the GE values quantified for the bacterium *B. ambifaria* were higher as compared to those of the other two bacterial strains, showing an increase during the three weeks after the inoculum in soil, in the presence of either AMF alone or AMF + biochar. The other two bacterial species may be less adaptable to non-optimal environmental conditions, with a reflection on growth rates. Effectively, data from the literature suggest that *B. amyloliquefaciens* has an optimal growth temperature between 30 and 40 °C, which is progressively reduced and stops around 15 °C [46]. *R. aquatilis* can tolerate harsh environments [47], but the dynamic growth for this and the other bacteria of MC_C must be explored further with specific studies. The data obtained from laboratory studies are, in fact, only suggestive of what may happen in a real environment in which many factors, like nitrogen source, kind of soil, and indigenous microbiota, can affect the microbial growth.

The growth trends did not seem to be different in the presence of the wheat cultivars Bramante or Svevo; it is likely that the plants did not significantly influence the colonization or the growth of the microorganisms. However, further analyses will be necessary to clarify whether there is a cultivar–microorganism interaction for the growth of the latter.

In general, the GE values were higher for all three PGPR strains in the presence of biochar, confirming its beneficial effect on bacterial persistence in the soil. Biochar can influence the structure and abundance of the soil microbiota by reducing nutrient leaching, altering plant–microorganism signaling pathways, and serving as a favorable growth habitat for their proliferation [19,34,48,49,50]. Biochar, in fact, possesses a porous structure suitable for internal and external colonization. Soil microorganisms can interact with the biochar surface ultrastructure through electrostatic or hydrophilic/hydrophobic interactions [19], while others establish themselves internally, lodging into the porous structure and finding an ideal environment sheltered from predators and other species competing for survival [48,49,50]. Similar experiments conducted in open fields and combining biochar and the commercial Micosat F1, a mix including both bacteria and fungi (CCS Aosta, Italy), have shown a positive effect of the same wheat cultivars employed in this study on grain yield and seed quality [34]. The positive effects were more evident with the combination of biochar and Micosat F1 than with the two supplemented separately, confirming the positive synergistic action of biochar and PGPM on plants [34]. Other studies have evidenced a positive effect of biochar combined with AMF, Micosat F1, or consortium MC_B [14] on the marketable production of tomato [51]. All these studies together confirm the positive synergistic action of biochar and PGPMs on plants, even though biochar’s properties and efficiency strictly depend on pyrolysis parameters such as temperature range and on the starting raw material [50].

This work has demonstrated that the developed qtPCR protocol allowed us to tag PGPR strains of consortium MC_C under field conditions, enlightening how these microorganisms not only remained viable in the soil up to one month after inoculation, but also were able to colonize and grow in the soil. This opens up the possibility of conducting specific studies to enrich the knowledge about plant/PGPR interactions and to explore the environmental factors affecting PGPR’s fate in the soil (nitrogen source, indigenous microbiota, kind of soil, etc.).

Of course, the protocol can be improved using strategies that allow for a more precise estimation of bacterial populations’ growth and that use genomic information to design primers. Despite the advantages offered by the qtPCR system, one of the main constrains remains in the impossibility of discriminating viable bacteria from dead ones [32]. By combining qtPCR analyses and staining with reagents like Propidium Monoazide, this limit can be overcome [52]. The results also suggest a positive effect of biochar on PGPR strains’ survival and growth, confirming all available data that biochar can bring several benefits to soil and to plants [19,34,53].

## Figures and Tables

**Figure 1 microorganisms-12-01002-f001:**
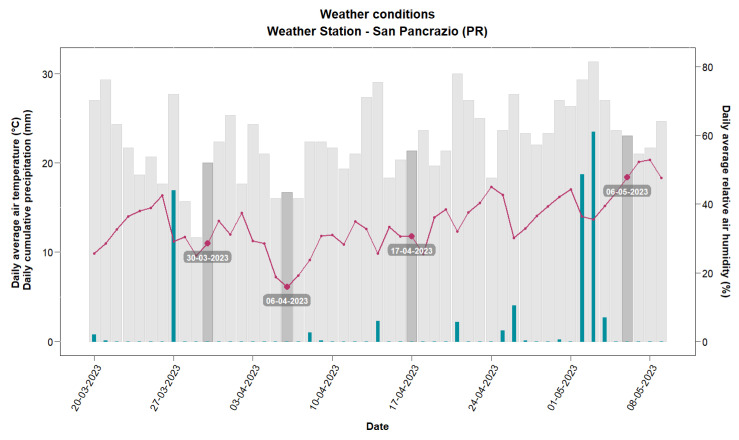
Weather parameters registered daily by an automatic computerized weather station installed close to the experimental field (San Pancrazio, Parma) during the sampling period (22 March 2023–5 May 2023). Grey histograms represent the daily average relative humidity (%); the red line, the daily average air temperature (°C); and blue histograms, the cumulative precipitation (mm). The darker grey histograms represent the dates of sampling.

**Figure 2 microorganisms-12-01002-f002:**
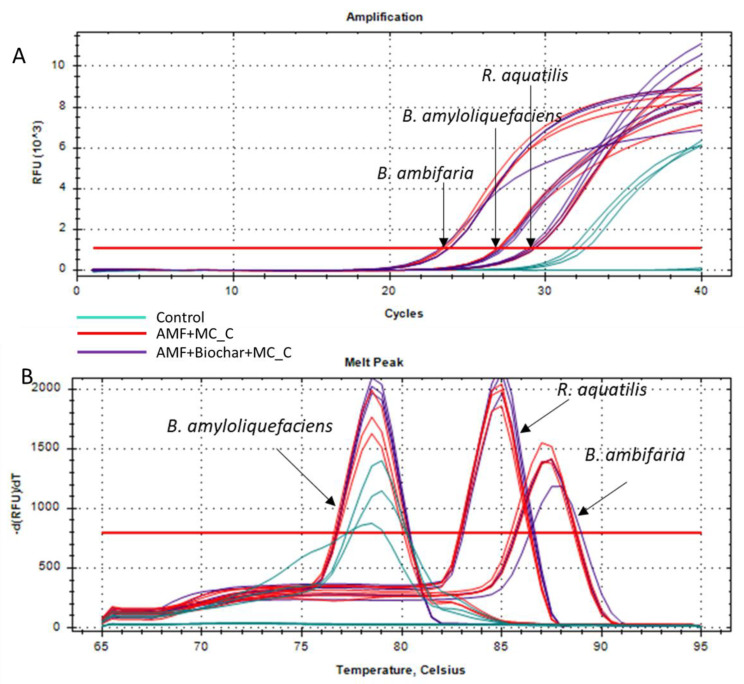
Test for the detection of *B. ambifaria*, *B. amyloliquefaciens,* and *R. aquatilis* in soil samples. Results of qtPCR analyses conducted on rhizospheric samples collected six days after sowing of the cv. Svevo seeds. (**A**) Amplification chart displays the relative fluorescence units (RFU) at every cycle. (**B**) Melting curves obtained based on amplicons’ dissociation conducted in the temperature range of 65–95 °C. Three replicates were performed for each sample. Horizontal red line represents the fluorescence threshold.

**Figure 3 microorganisms-12-01002-f003:**
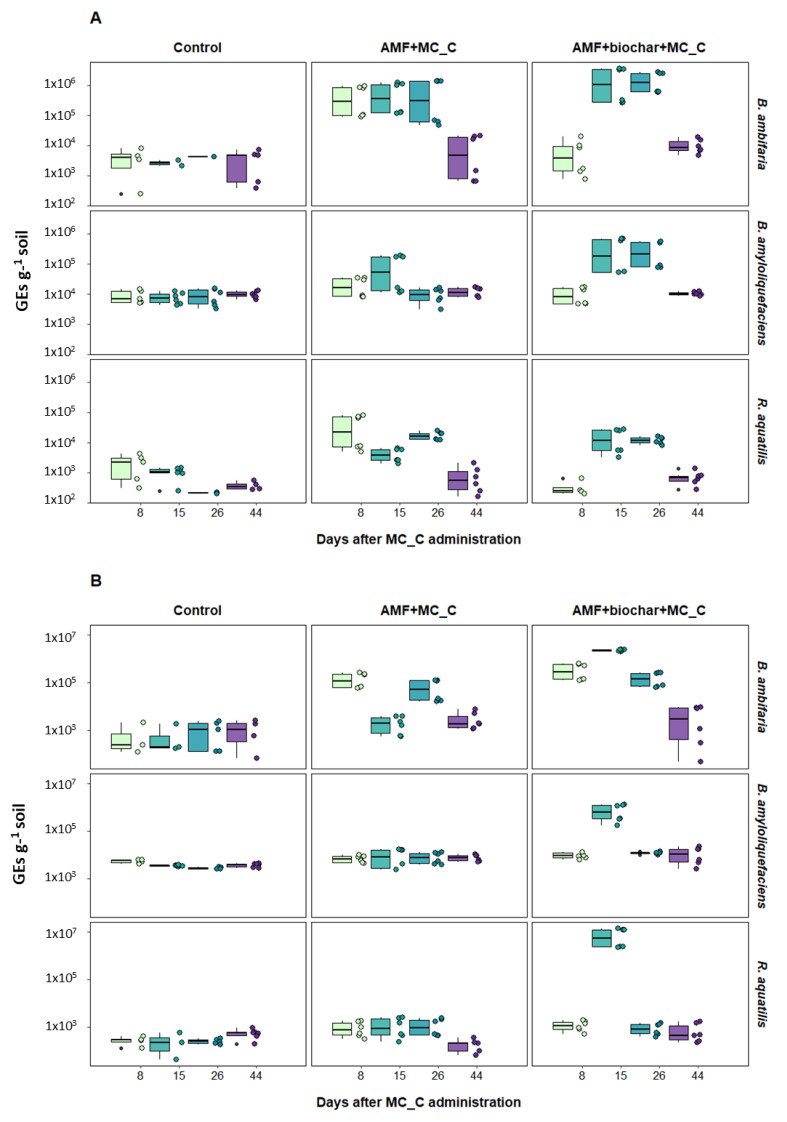
Quantification of GEs in the rhizospheric soil in the period between 30 March and 5 May, elaborated with R Studio software version 4.3.1. Boxplots are based on the quantification data (GEs) of all the qtPCR replicates derived by CFX Maestro^TM^ Software (Biorad) via interpolation with standard curves. Comparison of GEs values of *B. ambifaria*, *B. amyloliquefaciens,* and *R. aquatilis* for the three considered conditions: (i) control, without any treatments; (ii) AMF + MC_C; and (iii) AMF + biochar + MC_C for the wheat cv. Bramante (**A**) and cv. Svevo (**B**). Raw data are available in Appendix A.

**Table 1 microorganisms-12-01002-t001:** Bacterial species belonging to MC_C, developed by Tabacchioni et al. [14].

MC_C		
*Azotobacter chroococcum*	LS132	N-fixation
*Burkholderia ambifaria*	MCI7	Plant growth-promoting
*Bacillus* spp.	BV84	Biocontrol/plant growth-promoting
*Bacillus amyloliquefaciens*	LMG 9814	α-amylase, α-glucosidase, and iso-amylase production
*Pseudomonas fluorescens*	DR54	Biocontrol of pathogen
*Rahnella aquatilis*	BB23/T4d	Plant growth-promoting

**Table 2 microorganisms-12-01002-t002:** List of primer pairs selected for the analysis of strains of MC_C. The efficiency (E) and R^2^ were averaged on all standard curves (dilutions from 10^5^ to 10^1^ GEs) used for soil sample quantification. Tm: melting temperature of the amplicon; E: PCR efficiency, calculated as $E=10^{-1/\mathrm{slope}}-1$; R^2^: coefficient of determination; Nt: not tested. Reference genome was used to calculate GEs ng^−1^ DNA.

Target Species	Primer Name	Sequence	Tm	Reference	Reference Genome	E	R^2^
*R. aquatilis*	Raq4	Fw: 5′-CTCCAAACTGGTGCTGGAAG-3′ Rev: 5′-CAGCAGTTCCTGGGAGTTTG-3′	85.00 °C	UNIPR laboratory	ASM24195v1 (strain CIP 7865)	90.550 ± 6.650	0.942 ± 0.021
*B. amyloliquefaciens*	rpsj	Fw: 5′-ATCTGGTCCGATTCCGTTGCCG-3′ Rev: 5′-TGGTGTTGGGTTCACAATGTCG-3′	79.00 °C	[29]	ASM1939692v1 (strain GKT04)	93.750 ± 4.085	0.996 ± 0.003
*B. ambifaria*	Bamb 1196	Fw: 5′-CTGCGTTACACCGTCTTCG-3′ Rev: 5′-AAGTGGTCGCAATAGGCATC-3′	86.50 °C	[30]	ASM1612775v1 (strain FDAARGOS_1027)	Nt	Nt
	Bamb 3350	Fw: 5′-ACCCGTATCCAGCAGACCTT-3′ Rev: 5′-GTGCATGAACTCGACCGTCT-3′	87.00 °C			98.267 ± 9.128	0.996 ± 0.001
	Bamb 4475	Fw: 5′-CTACGTGAACCAGACGCTTG-3′ Rev: 5′-TCGACGAGTACGACGAGTTG-3′	87.50 °C			Nt	Nt

**Table 3 microorganisms-12-01002-t003:** *p* values obtained from pairwise comparison conducted by ANOVA. T2, T3, and T4 correspond to the samples collected at 15, 26, and 44 days after MC_C administration. ns = not significant (*p* > 0.05), * *p* ≤ 0.05, ** *p* ≤ 0.01, *** *p* ≤ 0.001, **** *p* ≤ 0.0001.

*B. ambifaria*-Bramante				
	AMF + MC_C All Times	AMF + Biochar + MC_C All Times	AMF + MC_C T2, T3, T4	AMF + Biochar + MC_C T2, T3, T4
CTR all times	2.92 × 10^−10^ ****	7.81 × 10^−9^ ****		
CTR T2, T3, T4			0.000350 ***	8.39 × 10^−6^ ****
AMF + MC_C all times		0.66514 ns		
AMF + MC_C T2, T3, T4				0.17311 ns
*B. ambifaria*-Svevo				
CTR all times	0.00367 **	0.00010 ***		
CTR T2, T3, T4			0.02886 *	0.00059 ***
AMF + MC_C all times		0.00730 **		
AMF + MC_C T2, T3, T4				0.01732 *
*B. amyloloquefaciens*-Bramante				
CTR all times	0.00998 **	0.00029 ***		
CTR T2, T3, T4			0.04268 *	1.21 × 10^−5^ ****
AMF + MC_C all times		0.05269 ns		
AMF+MC_C T2, T3, T4				0.00502 **
*B. amyloloquefaciens*-Svevo				
CTR all times	0.00170 **	2.70 × 10^−5^ ****		
CTR T2, T3, T4			0.00568 **	0.00014 ***
AMF + MC_C all times		0.00239 **		
AMF + MC_C T2, T3, T4				0.00553 **
*R. aquatilis*-Bramante				
CTR all times	0.00442 **	0.01267 *		
CTR T2, T3, T4			0.00028 **	0.00018 ***
AMF + MC_C all times		0.62860 ns		
AMF + MC_C T2, T3, T4				0.77247 ns
*R. aquatilis*-Svevo				
CTR all times	0.04419 *	0.00042 ***		
CTR T2, T3, T4			0.00226 **	0.00018 ***
AMF + MC_C all times		0.00247 **		
AMF + MC_C T2, T3, T4				0.00282 **

**Table 4 microorganisms-12-01002-t004:** Average and median ratio between bacterial GEs found in AMF + biochar + MC_C and AMF + MC_C samples. T2, T3, and T4 correspond to the samples collected at 15, 26, and 44 days after MC_C administration. Average: $\bar{X}=\sum_{i=1}^{n}x_{i}\cdot \;n^{-1}$. Median: value that occupies the central position in the data set.

*B. ambifaria*		
	Average	Median
Bramante all times	0.97107	1.04763
Bramante T2, T3, T4	1.10424	1.13713
Svevo all times	1.26138	1.34574
Svevo T2, T3, T4	1.31697	1.40049
*B. amyloliquefaciens*		
Bramante all times	1.09267	1.12214
Bramante T2, T3, T4	1.15601	1.15881
Svevo all times	1.12783	1.03542
Svevo T2, T3, T4	1.15195	1.00520
*R. aquatilis*		
Bramante all times	0.97500	0.96419
Bramante T2, T3, T4	1.01943	1.01178
Svevo all times	1.43474	1.15627
Svevo excluding T2	1.09839	1.13917
Svevo T2, T3, T4	1.56841	1.20323

## Data Availability

The original contributions presented in the study are included in the article, further inquiries can be directed to the corresponding authors.

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
