# Peer review of "Real-Time PCR (qtPCR) to Discover the Fate of Plant Growth-Promoting Rhizobacteria (PGPR) in Agricultural Soils"

_microorganisms, 2024, doi:10.3390/microorganisms12051002_

Round 1

Reviewer 1 Report

Comments and Suggestions for Authors

This paper had developed a Real Time PCR method with tracs and quantifies bacteria when added as microbial consortium. The results showed biochar had a positive effect on PGPR growth. And qtPCR was an effective method to trace the fate of supplied bacterial species, which is useful to discover the fate of PGPR in agricultrual soils.

Author Response

Thank you for the appreciation of our work. 

Reviewer 2 Report

Comments and Suggestions for Authors

Dear Authors,

In this paper, the authors utilize the real-time PCR technique to assess the growth of certain bacteria after inoculation into two varieties of wheat, studying the role of biochar addition. I think the article is well-written, the ideas are clear, the objective is also well-defined, and the experimental part is conducted correctly. However, I believe that some aspects need improvement and should be corrected.

Majors:

-L69: “(MC_A, MC_B and MC_C)” The meaning is not well understood, please explain it better.

--L85-89: I advise the authors to provide a better justification for why they consider qPCR to be the best method.

-L98: “2.1. Treatments” ? In reality, this section does not indicate treatments.

-L133: “were grown overnight in 2 mL of LB” Why were they allowed to grow for 2 days in LB medium, wouldn't it have been better without incubation? Is it to increase sensitivity? Could this have altered the results? Justify it. At what temperature were they grown?                                 

-Indicate in Figure 1 the 4 time points chosen for sampling in Figure 3. Also, there are too many time data points clustered together on the X-axis, making it unclear which column they correspond to.

-Table 1: Based on what criteria are those functions assigned to each bacterium in Table 1? Does each one only have one function? Can't they have several? What is indicated as the main one? Please provide references.

-The table 2 has numerous editing errors that need to be fixed.

-Caption of Figure 2: Include the number of repetitions done.

-L208: “3.2. Test on DNA Extracted from Soil” I advise the authors to use another title that better reflects the content of this section.

-L281: “To show that this difference did not rest solely on the high GEs found in the sample T2, a comparison was made excluding this sample”. I'm sorry, I don't understand what the authors are trying to say. Please rewrite.

-L282-L285: I don't understand those data or how the authors arrive at those conclusions. Where are the data (respectively 1.09839 and 1.13917) in table 4?

-In the discussion, a better explanation could be given for the reasons why PCR did not work with two out of the five bacteria of the MC-C, and what could be done about it.

L339: “The other two bacterial species may be less adaptable to non-optimal environmental conditions with reflection on growth rates” Is there any reference the authors can base this hypothesis on?

-In the discussion, I missed the authors discussing the role of nitrogen source present in the soil on the growth of these bacteria. Could the observed growth differences be due to this?

-L341-L344: The authors don't use a control of uncultivated soil, do they? So, what do they base this reasoning on?

-L350: “Soil microorganisms can adsorb to its surface”? This is poorly expressed, please rewrite.

-I miss in the discussion the authors giving an explanation of why they use the fungus Rhizophagus intraradices, in relation to the existing literature.

-The authors present meteorological data, but they don't use them in the discussion. What insights could these data provide?

Minors:

-L91: “to tag microorganisms of MC_C consortium” "The reason for choosing the MC_C is not clear. What organisms are they? Why these and not A or B?"

-L37: “….and fungi” I think microalgae also play an important role

-L59: “PGMPs”??

L102: zeolite. [15]. Typo

L148: “Varian Cary® 50 Bio UV-Visible Spectrophotometer” in bold? And also L238-L239

L144: “2.4. DNA Extraction” and quantification.

L153: “discovery” Find a better synonym

L169: ….. From there, there are several different font sizes

Round 2

Reviewer 2 Report

Comments and Suggestions for Authors

I believe the authors have adequately addressed all of my comments and suggestions, and I accept the paper in its current version

Author Response

Thank you for reviewing